# Genomic Analysis of Romanian Lycium Genotypes: Exploring *BODYGUARD* Genes for Stress Resistance Breeding

**DOI:** 10.3390/ijms25042130

**Published:** 2024-02-09

**Authors:** Roxana Ciceoi, Adrian Asanica, Vasilica Luchian, Mihaela Iordachescu

**Affiliations:** 1Research Center for Studies of Food Quality and Agricultural Products, University of Agronomic Sciences and Veterinary Medicine of Bucharest, 59, Mărăști Bd., 011464 Bucharest, Romania; roxana.ciceoi@qlab.usamv.ro; 2Faculty of Horticulture, University of Agronomic Sciences and Veterinary Medicine of Bucharest, 59, Mărăști Bd., 011464 Bucharest, Romania; adrian.asanica@horticultura-bucuresti.ro (A.A.); vasi_botanica@yahoo.com (V.L.)

**Keywords:** *BODYGUARD* genes, cuticle, goji berry breeding, plant resistance, whole genome sequencing

## Abstract

Goji berries, long valued in Traditional Chinese Medicine and Asian cuisine for their wide range of medicinal benefits, are now considered a ‘superfruit’ and functional food worldwide. Because of growing demand, Europe and North America are increasing their goji berry production, using goji berry varieties that are not originally from these regions. European breeding programs are focusing on producing *Lycium* varieties adapted to local conditions and market demands. By 2023, seven varieties of goji berries were successfully registered in Romania, developed using germplasm that originated from sources outside the country. A broader project focused on goji berry breeding was initiated in 2014 at USAMV Bucharest. In the present research, five cultivated and three wild *L. barbarum* genotypes were compared to analyse genetic variation at the whole genome level. In addition, a case study presents the differences in the genomic coding sequences of *BODYGUARD* (*BDG*) *3* and *4* genes from chromosomes 4, 8, and 9, which are involved in cuticle-related resistance. All three *BDG* genes show distinctive differences between the cultivated and wild-type genotypes at the SNP level. In the *BDG 4* gene located on chromosome 8, 69% of SNPs differentiate the wild from the cultivated genotypes, while in *BDG 3* on chromosome 4, 64% of SNPs could tell the difference between the wild and cultivated goji berry. The research also uncovered significant SNP and InDel differences between cultivated and wild genotypes, in the entire genome, providing crucial insights for goji berry breeders to support the development of goji berry cultivation in Romania.

## 1. Introduction

Goji berry plants have long been used for both Asian culinary and medicinal traditions, with their use extending back over thousands of years [1], and, currently, the berries are acknowledged as one of the most recognised ‘superfruits’ of the 21st century [2,3,4], being considered as a functional food [1,5]. The goji berry has attracted significant attention in Western countries due to its nutritional profile, especially for its abundant vitamins and antioxidants. Its oxygen radical absorbance capacity values, which lie between 25,000 and 30,000, surpass those of other nutritionally beneficial fruits like pomegranates and blueberries, indicating its superior antioxidant capacity [6]. Its medicinal uses range from improving visual acuity [5,7,8], abdominal pain [5], dry cough, fatigue and headache [5], immune system support, cancer prevention [7,8], and antidiabetic activity [7,8] to increased longevity [8,9,10] and enhanced fertility [10,11,12,13].

In China, out of the existing nine *Lycium* taxa [14], only four are traditionally utilised, with *L. barbarum* and *L. chinense* being the main species traded worldwide [14,15]. In World Flora Online, the genus *Lycium* comprises 436 species names, and out of which 92 are accepted species, 241 are considered synonyms, and 103 are unplaced [16]. Yao and al. name 97 *Lycium* species, and out of which 35 species and 2 varieties are used as food and/or medicine worldwide [14]. The Plants of the World Online platform includes 101 officially accepted *Lycium* species in 71 countries, across 130 regions, including Romania [17]. Such taxonomical debate could also be explained by the fact that the genetic foundation of the germplasm resources of wild *Lycium* species in the world, and also in China, remains poorly understood [18].

The flora of Romania recognised *Lycium halimifolium* L. as a native species for decades [19] before *L. barbarum* became the accepted name [17,20]. A manuscript from 1867 documents the traditional usage of *Lycium vulgare* Dun. in Romania, and mention its identification as *L. barbarum* in the Transylvania region [21,22]. Although *L. halimifolium* is a synonym of *L. barbarum* [20,23,24], widespread public belief still treats *L. halimifolium* and *L. barbarum* as separate species, attributing them different culinary and toxicological properties [25,26]. Traditionally, the plant has been used extensively to make fences in the countryside, but has also had folk medicinal uses, such as in treating conditions related to fear and anxiety and for epilepsy and spasms, indicating psychological and neurological benefits [27]. In a few Romanian regions, it is considered an invasive plant, such as in Oltenia, the Danube riverbanks, and Dobrogea [28,29].

Due to goji berries’ increased fame, the market demand has grown exponentially in the last two decades [30,31]. China dominates goji berry production, particularly in the northwest regions like Ningxia and Xinjiang, the two main exporting regions [9,32,33]. In contrast, production in North America and Europe is limited due to a lack of traditional use, knowledge, and adapted varieties [34,35,36,37]. Romania has emerged as a significant producer of goji berries [38], also focusing on plant material for cultivation [39], with a market that is showing a rising trend [40]. Especially in the difficult context of climate change constraints, goji berry planting material which has adapted to local conditions is required by European farmers. Therefore, *Lycium* breeding programs have been launched, together with initiatives on identifying promising genitors and new crop production processes [41].

By 2023, seven varieties of goji berry had been registered in the Official Catalogue of Cultivated Plant Varieties: ‘Erma’, ‘Transilvania’, ‘Kirubi’, ‘Kronstadt’, ‘Bucur’, ‘Sara’, and ‘Anto’, belonging to both *L. barbarum* and *L. chinense* [42].

Having a deeper understanding of native goji berry genetic resources is important both for preserving local biodiversity and for the breeding sector [18,33,43]. With growing market demand for goji berries, comprehensive molecular research has been initiated to identify valuable genes in both cultivated and wild goji berry plants, aiming to enhance future breeding programs [1,33,43,44,45,46]. Crop breeding aims to develop new plant varieties with improved traits such as increased yield, disease resistance, and nutritional quality [47]. High-throughput technologies, including genomics, transcriptomics, and metabolomics, have opened up a new phase in crop breeding, enhancing the efficiency and precision of this process [47,48]. The last two decades have seen a significant growth in both the volume and quality of publicly available plant genomes, with a higher efficiency of genome sequencing, assembly, and annotation [48,49,50].

In the Solanaceae family, which includes around 3000 species, 170 full genomes of 46 species have been sequenced [49]. Among them are *Lycium barbarum* [47,51] and its invasive relative, *L. ferocissimum* [52]. The *L. barbarum* genome contains 12 chromosomes [31] (2n = 2x = 24) and it is 1.8 Gb in size, with a level of heterozygosity of approximately 1% [51]. The sequenced and annotated genome ASM1917538v2 [53] was obtained by sequencing a haploid plant developed from pollen culture, using PacBio Sequel technology [51]. The annotation allowed for the identification of 47,740 genes and 34,339 protein-coding sequences. The availability of another annotated genome of *L. ferocissimum*, of 1.2 Gb size, 40,291 genes, and 30,549 protein-coding genes [52], will ease the characterisation of the future goji berry sequenced genomes even more, allowing for the identification of new genes of interest.

The current research marks the initial phase of a broader project focused on genes related to resistance to abiotic and biotic stress. The present study is a preliminary exploratory step that aimed to discover regions with high SNP and InDel polymorphism as sources of wild-type resistance genes that could be introgressed into future varieties. A case study on the genomic coding sequence of *BDG* genes, focusing on cuticle thickness, is presented to demonstrate the utility of the research. Analysing the genetic diversity of cultivated and wild goji plant genes has the final aim of providing information required by goji berry breeders, supporting the development of goji berry production in Romania.

## 2. Results

### 2.1. NGS Data Analysis

#### 2.1.1. Sequencing Data Quality Control

The genomes of eight Romanian *L. barbarum* varieties, out of which five were cultivated varieties that were part of a population obtained from Chinese seeds [54] and three were spontaneous plants growing in the wild in three different Romanian counties [28,55], were sequenced using NGS technology. The distribution of sequencing quality was analysed across the entire length of all sequences to identify any locations with abnormally low sequencing quality that could indicate the inclusion of incorrect bases at higher-than-normal rates. Novogene Co., Ltd. (Cambridge, UK), analysing base calling (Casava 1.8 software), had Qphred scores between 30 and 40, indicating error rates between 1:1000 and 1:10,000, with the Qphred usually being higher than 35 (Appendix A, Sequencing Quality Distribution). The sequencing error rate for all samples was around 0.02 at the beginning of the data acquisition and between 0.04 and 0.06 at the end of the reading (Appendix A, Sequencing Error Rate). When performing sequencing data filtration, the percentage of clean reads was between 99.52% and 99.72% (Appendix A, Classification of the Sequenced Reads). Regarding the statistics of the sequencing data, for 1,669,720,889 base pair (bp) reference genome, the mapping rate of each sample ranged from 96.66% to 99.36% (Appendix A, CleanData_QCsummary). The proportion of clean data relative to raw data, referred to as the effective rate, was higher than 99.52% for all reads. Referring to the reference genome (without Ns), the average depths were between 10.01 X and 9.29 X and the 1 X coverages ranged from 77.43% to 97.41%; the results therefore fell within the acceptable normal range and could be utilised in variation detection and genetic analysis (Appendix A, Allsample_allinfo).

#### 2.1.2. SNP Detection, Distribution, and Mutation Frequency

SNP (Single Nucleotide Polymorphism) variations were observed in all eight genotypes, but the quantity and genomic distribution of these variations differed across the genotypes. A total of 108,290,958 SNPs were identified within the eight genotypes, with an average ranging from 14,079,300.6 SNPs/genome for the cultivated specimens and 12,631,485 SNPs/genome for the wild specimens. The Lb2 genome exhibited the largest quantity of SNPs, totalling 15,983,773.

In the eight genotypes, the transitions—point mutations that change one purine nucleotide to another or one pyrimidine to another—were more frequent, with an average count of 8,559,837.5. This was higher than the number of transversions, which are mutations that switch a purine for a pyrimidine or vice versa, averaging at 4,976,532.25. This resulted in an average ts/tv ratio (transitions to transversions) of 1.72, which was relatively consistent across the two categories of genotypes, cultivated and wild. However, there was a difference between the cultivated and wild-grown plants: the average ts/tv ratio was higher in the cultivated genotypes at 1.736, compared to 1.688 in the wild-grown plants.

The genotype Lb7w exhibited the highest heterozygosity rate, measured as 5.037‰ (per thousand), whereas the lowest rate was found in genotype Lb4, at 3.245‰. On average, the wild plants showed a higher heterozygosity rate, averaging at 4.851‰, compared to the cultivated plants, which had an average heterozygosity rate of 3.607‰.

Figure 1 illustrates the distribution of the six types of SNP mutations. Genotypes Lb1 and Lb2 showed the highest number of SNPs across all six SNP types, while genotype Lb8w had the lowest count. Among these six types of SNP mutations, for every genotype, the most common was the C:G>T:A mutation, followed by T:A>C:G. Conversely, the least frequent type of SNP was the C:G>G:C mutation. For the C:G>T:A mutation type, the Lb2 genotype had 5,289,157 SNPs, Lb1 had 5,256,956 SNPs, and LB8w had 3,784,249 SNPs. For the T:A>C:G, the Lb2 genotype had 4,877,268 SNPs, Lb1 had 4,854,020 SNPs, and LB8w had 3,768,683 SNPs. For the C:G>G:C, the Lb2 genotype had 934,963 SNPs, Lb1 had 931,895 SNPs, and LB8w had 731,795 SNPs (Appendix A, SNP.frequency and SNP_Annotation_Statistics).

#### 2.1.3. Insertion/Deletion Detection and Distribution

InDel (insertion and deletion) variations were detected in all the studied genotypes, amounting to a total of 11,225,960 InDels, averaging 1,403,245 InDels per genome. The genotype Lb2 had the highest number of InDels at 1,773,325, whereas Lb8w had the fewest, totalling 1,197,605. For the cultivated specimens, the average InDels per genome were 1,495,506.4, compared to 1,249,476 for the wild specimens. When examined individually, the overall count of insertions, which was 5,166,205, was less than the total number of deletions, totalling 6,043,098. However, the mean count of insertions in the cultivated specimens, at 684,984.8, exceeded the average insertion count in the wild specimens, which was 580,427. In the case of deletions, the average number of deletions in the cultivated specimens, amounting to 808,410.6, was greater than the average deletion counts in the wild specimens, recorded at 667,015.

The InDel heterozygosity rate, expressed in per mille (‰) and calculated as the ratio of InDels to the total genomic bases, was 0.381‰. This value was lower on average for the cultivated specimens, at 0.335‰, compared to the higher average rate for the wild specimens, which was 0.458‰.

InDel distribution within the genome (Figure 2) showed that almost 50% of all insertions and deletions (InDels) had the length of 1 base pair, around 13% of InDels were 2 bp long, around 7% of InDels were 3 bp long, and, thereafter, the percentage continued to decrease with the increase in InDel length. The highest percentages of 1 bp InDels were observed in the Lb5 (49.40%), Lb4 (49.32), and Lb3 (49.28%) cultivated genotypes, while the lowest percentages were observed in the wild genotypes Lb8w (48.42%), Lb6w (48.32%), and Lb7w (48.21%). On the contrary, the highest percentages of 2 bp InDels were observed in the wild genotypes Lb7w (14.12%), Lb6w (14.11%), and Lb8w (14.05%), while the lowest percentages were observed in the Lb1 and Lb2 genotypes (13.29%). The InDels longer than 12 bp were below 1%, and the ones longer than 32 bp were below 0.1% (Appendix A, InDel.GENOME percentage and InDel_Annotation_Statistics).

In the analysis of the eight genotypes, the densities of SNPs (Figure 3) and InDels (Figure 4) across each chromosome appeared to be relatively similar (Appendix A). However, a visible reduction in InDel density was observed (Figure 3). Additionally, distinct differences were evident between the genomes of the cultivated and wild plants. Generally, the ratio of SNPs to InDels was around 10. This ratio was slightly lower in the cultivated plants, ranging from 9.29 to 9.52, and was somewhat higher in the wild plants, with values ranging from 10.03 to 10.16.

For both SNPs and InDels, it was visible that the density of variation was higher at the end of all 12 chromosomes. In addition, (1) all genomes of the cultivated specimens had high variation density, for both SNPs and InDels, almost in the middle of the 5th chromosome, with the same area also being observed in the genomes of the wild plants, but with a lower density; (2) all genomes had the first half of the 12th chromosome with a very low density of both SNPs and InDels, with the exception of the wild plants, where SNPs were present at a very high density; (3) the 1st chromosome, despite being the longest one, had the longest area with a low density of SNP and InDel variations, except for Lb8w; (4) there was a clear distinction between the cultivated and wild plant genomes, which was more easily observed at the level of SNPs.

#### 2.1.4. Sequence Analyses of *BODYGUARD* Genes in Romanian Goji Berry Genomes

In both *Lycium* species with available reference genomes, three *BODYGUARD* (*BDG*) genes were identified (Table 1). However, these genes are situated on different chromosomes in each species, on chromosomes 4, 8, and 9 in *Lycium barbarum* and on chromosomes 1, 3, and 9 on *Lycium ferocissimum*.

Analysing the eight studied genomes using Genome Workbench, a distinct divergence was noted between the genomes of the cultivated and wild plants.

Regarding the *BDG* gene situated at LOC132634709 (Table 2) on chromosome 4, 22 SNPs were identified within its coding region. Out of these, 14 are synonymous mutations, meaning they do not change the amino acid sequence. With the exception of the SNP at position 1312, situated within a codon that encodes for lysine/arginine (basic amino acids), the rest of the SNPs are situated within codons that encode for either nonpolar (10) or polar amino acids (11), and none of these SNPs change the polarity or charge of the encoded amino acid. A key finding was the apparent distinction in most SNPs between the cultivated and wild varieties of the plant, with differences being observed as homozygous versus heterozygous SNPs.

In the *BDG* gene at LOC132607278 (Table 3) on chromosome 8, the analysis revealed 28 SNPs within its coding sequence, including 4 synonymous mutations. Notably, SNPs at positions 287–288 and 392–294 each impact a single codon, changing AAA to CGA (Lysine to Arginine) and AAA to GAC (Lysine to Aspartic Acid), respectively. A distinct pattern was observed between the cultivated and wild genotypes at 17 specific SNP locations: 174, 209, 288, 392–394, 617, 648, 688, 694, 799, 891, 1030, 1049, 1076, 1150, 1151, 1266, and 1304, predominantly presenting as homozygous versus heterozygous SNPs. Apart from the SNPs at positions 146 and 1274, which alter the charge of the encoded amino acids (Lysine to Glutamic Acid and Glutamic Acid to Lysine, respectively) from basic to acidic and vice versa, the other SNPs do not cause any changes in either the polarity or charge of the encoded amino acids.

The *BDG* gene at LOC132609965 (Table 4) on chromosome 9 features 29 SNPs within its coding region, with 18 of these being silent mutations. Two SNPs at positions 555–557 lead to the formation of four different codons: TCT, GCT, TCA, and TCG, which correspond to the amino acids Serine, Alanine, Serine, and Serine, respectively. In a similar manner, the two SNPs at positions 1671–1672 result in the codons GGA, AGA, and GAA, which encode for the amino acids Glycine, Arginine, and Glutamic Acid, respectively. The SNPs at positions 555–557 are unique in that they alter the polarity of the encoded amino acid from Serine (as found in the reference genome) to Alanine (as observed in the cultivated genotypes). The other SNPs, however, do not cause any changes in the polarity or charge of the amino acids that they encode. Approximately half of these SNPs show sequence-level differences between the cultivated and wild-type genotypes.

## 3. Discussion

Exploring the genetic diversity in Romanian wild and cultivated *Lycium* species can advance breeding by developing new varieties tailored to specific environmental conditions and market needs, highlighting key genetic markers for desired traits. The Romanian homologated varieties were developed based on Chinese varieties’ germplasm, due to their high fruit quality traits [56,57], without using the local germplasm. Generally, the goji berry in Romania only has three major biotic threats, powdery mildew, goji berry gall mite, and stink bugs [58,59], making it much more suitable for organic production than that in China [37,60]. The escalating threat of extreme weather events caused by climate change is set to pose an increasingly serious challenge to goji berry production, with the major threats being extreme drought and insolation [61,62]. The cuticle is a protective, hydrophobic layer covering the epidermis of leaves, stems, and fruits in plants [63,64]. Cuticle primary roles include water regulation, protection against biotic stress, defence against abiotic stress, and the facilitation of gas exchange and photosynthesis, enhancing pollution tolerance [65,66,67,68,69,70]. The cuticle type also impacts the fruits’ postharvest storage [71], as demonstrated for the goji berry [67]. By 2013, Yeats and Rose had mentioned almost 50 discovered cuticle-associated genes, with most of them belonging to Arabidopsis, tomato, rice, barrel clover, and maize [63]. Among these are *BODYGUARD* genes that encode proteins like α/β hydrolase, crucial for plant defence and cutin biosynthesis in Arabidopsis [67,72,73,74]. Studies on goji berry cuticles have revealed that certain varieties have enhanced resistance to *Alternaria alternata*. This offers valuable preliminary data for breeding and selecting cultivars for better postharvest storage [64].

The sequencing and annotation project of the goji berry genome in 2023 [51] represents a crucial resource for future resequencing projects. The advancements in next-generation sequencing/whole genome sequencing (NGS/WGS) [50] are poised to generate a wealth of data, which will be instrumental in developing new goji berry varieties.

Following SNP and InDel density analysis, it became apparent that SNP polymorphism is more spread from the chromosome ends towards their middle part, whereas InDel polymorphism is more concentrated in the chromosomes’ ends. In addition, in the beginning of chromosome 12, there is much less InDel polymorphism compared to the rest of the chromosome ends (Figure 3 and Figure 4). Higher SNP and InDel densities have been observed in other plant species such as *Sorghum* spp. [75,76], *Solanum lycopersicum* L. [77,78], and *Capsicum* spp. [79]. The observed increased polymorphism near the ends of chromosomes can be attributed to the higher frequency of recombination in these areas [80]. In addition to chromosome ends, the wild-type genotypes Lb6w and Lb7w present a high degree of SNP polymorphism in the central regions of chromosomes 9 and 10 (Figure 3). The examination of variations in density at the genomic level, particularly for SNPs and InDels, highlights specific genome areas that warrant further investigation, to identify potentially beneficial genes from wild genotypes that could be integrated into new varieties.

Romanian breeding efforts have led to the registration of seven new goji berry varieties in the Official Catalogue of Cultivated Plants in Romania. These include ‘Erma’ and ‘Transilvania’ registered in 2017, ‘Kirubi’ in 2018, ‘Kronstadt’ in 2019, ‘Bucur’ and ‘Sara’ in 2020, and ‘Anto’ in 2021 [40]. This development has enabled Romanian farmers to establish commercial *L. barbarum* and *L. chinense* plantations using certified plants. Presently, commercial plantations and branded products are established in several Romanian counties, including Bihor, Brașov, Călărași, Cluj, Constanța, Dâmbovița, Hunedoara, Prahova, Satu Mare, Sibiu, and Vaslui, and this trend is on the rise, so new varieties are being requested by the market. Present research is dedicated to enriching the diverse gene pool found in wild germplasm, potentially enhancing the unique characteristics of Romanian goji berries. By examining the morphological and phenological traits of wild goji berries and correlating them with genetic data, characteristics like early or late flowering, high drought tolerance, and strong resistance to low temperatures, as well as features like thicker cuticles and leaves, could become valuable assets in breeding programs.

In earlier research, the morpho-anatomical features of leaves and flowers of both wild and cultivated goji berries in the Bucharest region were analysed. One study aimed to identify the key traits of interest to both goji berry breeders and taxonomists [28]. Another study involved mapping the spontaneous genetic resources found across Romania [55]. Notable morphological distinctions were observed in the leaf shape, orientation, and width of Romanian *L. barbarum*, results that are similar with findings reported in the Republic of Moldova, in a similar study between cultivated and wild goji berries [81]. Leaf anatomical characteristics are particularly significant in relation to biotic and abiotic stress factors, with wild plants having leaves covered with a thick cuticle, prominently developed vascular bundles, and sheaths surrounding the vascular bundles within the mesophyll. Additionally, the palisade cells in these plants were observed to be considerably larger than those in the cultivated plants [28]. These findings motivate further investigation into genes putatively linked to these phenotypic differences. The formation of the plant cuticle involves several proteins that play crucial roles in the biosynthesis and regulation of cutin and waxes, such as BDG, CER, KCS, VLCFAs, GPAT, LACS, ABC, SHN/WIN, LTPs, and CD1 [63,67,72,82].

Arabidopsis *BDG1* proved to be involved in multiple processes: cuticle development [67,72,83,84], cutin biosynthesis and response to osmotic stress [85], defence response to the fungus Botrytis cinerea [83], lateral root development [84], the positive regulation of cutin biosynthesis, suberin biosynthesis, and transpiration [67]. All of these studies used the Arabidopsis *bdg* mutant phenotype to prove BDG1’s functions. For instance, *bdg* mutant plants are dwarfed and have abnormal leaves, collapsed cells, a reduced number of trichomes, and an abnormal cuticle, as they accumulate more cell-wall-bound lipids and epicuticular waxes than wild-type plants and have activated defence responses, making them immune to *Botrytis cinerea* attack [72,83]. However, *bdg* mutant plants are extremely sensitive to osmotic stress [85].

Three *BDG* genes, similar to Arabidopsis *BDG 1* which encodes a protein involved in cutin biosynthesis and cuticle development and morphogenesis [69,71], were selected for a detailed analysis. In the reference genome, the *BDG* genes are located in high-SNP and -InDel polymorphism areas. The gene LOC132634709, a probable lysophospholipase *BODYGUARD 3*, is located at the beginning of chromosome 4, position 426077–430655. The gene LOC132607278, a probable lysophospholipase *BODYGUARD 4*, is located at the end of chromosome 8, position 127610658–127620151. The gene LOC132609965, a probable lysophospholipase *BODYGUARD 3*, is located at the end of chromosome 9, position 126096652–126103895.

In analysing the sequences of three *BODYGUARD* (*BDG*) genes in Romanian goji berry genomes, notable differences between the cultivated and wild types are evident, as observed in Table 2, Table 3 and Table 4. For the *BDG 3* gene on chromosome 4, 14 out of 22 SNPs (64%) can distinguish wild from cultivated types. Eight SNPs (positions 590, 749, 1208, 1247, 1352, 1439, 1715, and 1886), all of them silent, do not differentiate between cultivated and wild types. On chromosome 8’s *BDG 4* gene, 17 out of 26 SNPs (69%) do so. For chromosome 9’s *BDG 3* gene, 15 out of 28 SNPs (56%) differentiate between the two types. Seven of the thirteen SNPs that do not differentiate between the cultivated and the wild-type genotypes are silent. These findings highlight significant genetic variations between cultivated and wild goji berry plants. It is not yet certain how each of these SNP variations at the gene sequence level translates into phenotypical differences between the wild-type and cultivated plants. In Arabidopsis, the use of loss-of-function mutant plants obtained by transposon insertion led to the discovery of BDG1 multiple roles. Previous studies demonstrated morphological differences between wild and cultivated goji berry plants [55,81]. It remains to be seen in future studies if these differences in morphology are directly linked to the gene sequence variations, if indeed the wild plants are resistant to various pathogens, and to what extent they are affected by abiotic factors, such as osmotic stress [85].

The sequence analyses of the *BDG* genes in Romanian goji berry genomes revealed several differences among the three genes. The genes located on chromosomes 4 and 9 encode probable lysophospholipase BODYGUARD 3 proteins, whereas the gene located on chromosome 8 encodes a probable lysophospholipase BODYGUARD 4 protein [51]. The *BDG 4* gene from chromosome 8 is shorter than the *BDG 3* genes from chromosomes 4 and 9. Although located on different chromosomes, two of the genes presented SNPs affecting the same amino acid, such as in the 12, 65, 84, 235, 254, 410, 426, 467, and 473 positions. Even if some SNPs are located within conserved regions, many of them are silent (Figure 5).

Recent advances in genetic research have significantly enhanced our understanding of both goji berries and other important crop species. Regarding the goji berry, a comprehensive analysis of the relationships and origins of various *Lycium* species, including wild and cultivated varieties in China, was proposed by Qian et al. [86], while quantitative trait loci for fruit size in goji berries, employing specific-locus amplified fragment sequencing for SNP detection, were determined by Rehman et al. [87]. For soybean, genome resequencing and the development of SNP markers provided a framework that could be adapted for goji berry genetic studies and breeding [88,89]. For groundnut, Pandey et al. developed a high-density SNP array, a technique that can also be applied to goji berries to explore genetic diversity [90]. In tomatoes and apples, two teams demonstrated the utility of genomic libraries and reduced representation genome sequencing offering valuable methods that could be employed in goji berry genetic research [91,92]. These studies collectively indicate a growing trend of employing advanced genomic techniques to enhance crop breeding and genetic analysis, with potential applications in understanding and improving goji berries.

## 4. Materials and Methods

### 4.1. Plant Material

In this study, five selected Romanian-goji-berry-cultivated and three spontaneous-growing genotypes were examined. The five genotypes are integral to an extensive breeding program for goji berries that commenced in 2014 at the Experimental Field of the Faculty of Horticulture, at the University of Agronomic Sciences and Veterinary Medicine in Bucharest [54,56,57]. The initial biological samples were derived from the seeds of *Lycium barbarum* L., including five distinct biotypes: Lb1–Lb5 [54]. The native plant samples were chosen from robust and well-established populations in the counties of Bucharest (Lb6w), Ilfov (Lb7w), and Călărași (Lb8w). Specifically in Bucharest, specimens were gathered from the shores of Morii Lake (44.453424, 26.013337), a natural area on the periphery of the western segment of the Romanian capital. This location was also selected for a comparative morpho-anatomical study of the leaves and flowers of both wild and cultivated goji berry plants [28]. The plants encountered in Ilfov county are believed to have originated from cultivated specimens within a military base, subsequently becoming naturalised in the area (44.447382, 26.019239). The specimens from Călărași were found to be proliferating along a roadside in Lehliu city (44.434389, 26.858775). Voucher specimens for all eight genotypes were stored in the Herbarium BUAG “Prof. dr. V. Ciocîrlan” of USAMV Bucharest, entry numbers 4094–4101 (Appendix A, Sampling Metadata Sheet).

### 4.2. DNA Extraction

Genomic DNA from fresh goji berry leaves was isolated using the InnuPure C16 automated system (Analytik Jena GmbH, Jena, Germany), which employs magnetic particle separation technology for the fully automated extraction and purification of DNA. This process took place at the Research Center for Studies of Food Quality and Agricultural Products at the University of Agronomic Sciences and Veterinary Medicine in Bucharest, Romania. For genomic DNA extraction, the InnuPREP Plant DNA I Kit-IPC16 (Analytik Jena GmbH, Jena, Germany) was used, adhering to the protocols provided by the manufacturer. Initial processing involved breaking down the plant material externally, with the sample being mashed into a fine powder under liquid nitrogen and then homogenised using an SLS lysis solution (with CTAB as the detergent), proteinase K, and an RNase A solution. Following this external lysis step, the automatic DNA extraction continued in the InnuPure C16 automated system, as per the manufacturer’s guidelines. DNA quantification was performed using a NanoDrop™ 1000 spectrophotometer (Thermo Fisher Scientific, Wilmington, DE, USA) [78].

### 4.3. Sequencing and Sequencing Data Quality Control

Whole genome sequencing (WGS) was conducted using the next-generation sequencing (NGS) technology of an Illumina platform by Novogene Co., Ltd. (Cambridge, UK). The original image data from Illumina’s high-throughput sequencing were converted into sequenced reads (raw data) through the CASAVA base recognition process (Base Calling) at Novogene Co., Ltd. These raw data were saved in FASTQ (.fq) format files [93], which included the sequencing reads along with their respective base quality scores. In NGS, as different factors (choice of sequencing platform, chemical reactants, sample quality, etc.) can influence the overall sequencing quality and the rate of base errors, an assessment across the entire length of all sequences was performed. This process allowed for the identification of specific sites or base positions that exhibited unusually low sequencing quality, which translated into high levels of incorrect base incorporation. When using Illumina platforms, the error rate of sequencing is denoted by ‘e’. The quality of sequencing, referred to as Q_phred_, is a score assigned to each base (Phred score) to indicate its accuracy. This Phred score is calculated using the following formula: Q_phred_ = −10 log10(e). Essentially, this formula translates the sequencing error rate into a quality score [94]. Lower Q scores are associated with a rise in false-positive variant calls, which can lead to erroneous conclusions and additional costs for confirmatory experiments. Illumina’s sequencing technology consistently achieves Q30 or higher scores for most bases. This level of precision is particularly beneficial for various sequencing applications, including those in clinical research, where reliable data are critical [95]. In addition to sequencing quality distribution, on Illumina high-throughput sequencing platforms, the error rate has to be determined, as this increases with read extension, due to the consumption of chemical reagents during the sequencing process. Sequencing data filtration involves cleaning raw sequencing reads to enhance downstream analysis quality. This process includes removing paired reads if either contains adapter contamination, discarding paired reads where uncertain nucleotides (Ns) exceed 10% of either read, and eliminating paired reads with more than 50% low-quality nucleotides (base quality ≤ 5). All of this was performed by Novogene Co., Ltd. (Cambridge, UK) and results are provided as statistics in a sequencing data table.

### 4.4. Computational Data Processing and Sequencing Analysis

BWA software was utilised to align the effective sequencing data with the reference sequence, using the following parameters: mem -t 4 -k 32 -M [96]. The alignment outcomes were used to calculate the mapping rate and coverage.

The reads were aligned with the reference genome of the goji berry, GCF_019175385.1, downloaded from the NCBI database [51]. This process produced sequence alignment format files, which were subsequently transformed into binary sequence alignment format (*.bam) files. These were then processed to generate a variant file containing SNP (Single Nucleotide Polymorphism) data. The mapping rates for the samples indicate the degree of resemblance between each sample and the reference genome. Additionally, depth and coverage serve as metrics for the consistency and extent of correspondence to the reference genome, as conducted by Novogene Co., Ltd.

### 4.5. SNP Detection and Annotation

SNP detection was conducted using SAMtools with the specified parameter ‘mpileup -m 2 -F 0.002 -d 1000’ [96], facilitated by Novogene Co., Ltd. To minimise the likelihood of errors in SNP identification, the data underwent a two-step filtration process: an SNP was only considered if it was supported by over four reads, and its mapping quality had to exhibit a root mean square value exceeding 20, based on the supporting reads’ mapping qualities. The overall heterozygosity rate of SNPs across the genome (het. rate, denoted in permille ‰) was determined by the number of heterozygous SNPs over the total count of genomic bases. SNPs were assorted into six mutation classifications: T:A>C:G, T:A>G:C, C:G>T:A, C:G>A:T, T:A>A:T, and C:G>G:C. Take, for instance, mutations from T:A to C:G, which entail alterations from T to C and A to G. A T-to-C mutation on one strand of the DNA double helix will correspond to an A-to-G mutation at the identical position on the opposite strand. As a result, mutations of T>C and A>G were grouped together into one category.

### 4.6. Insertion/Deletion (InDel) Detection and Annotation

An InDel was identified as either an insertion or a deletion of a DNA sequence that is 50 base pairs (bp) in length or shorter. The detection of InDels was carried out using SAMTOOLS with the parameter set to ‘mpileup -m 2 -F 0.002 -d 1000′ [96], annotated with ANNOVAR software [97], by Novogene Co. The criteria for filtering InDels to enhance detection accuracy were consistent with those applied during SNP detection. The length distribution of InDels was examined as a proportion of the entire genome.

### 4.7. Sequence Analysis of the BDG Genes

Next-generation-sequenced BAM files containing the nucleotide sequence data for the eight goji berry genotypes were uploaded onto NCBI genome Workbench software, version 3.9.0, and aligned to the reference genome [98]. For each variety, the differences in nucleotide sequence were noted. Amino acid sequences of the three probable BDG proteins were aligned using MultAlin software, version 5.4.1 [99].

## 5. Conclusions

The present study re-sequenced the whole genome for eight *L. barbarum* genotypes, both cultivated and wild-type, and analysed the variability of three *BDG* genes, involved in cuticle biosynthesis, at the coding sequence level. NGS sequencing revealed clear differences between the cultivated and wild-type genotypes, not only in the whole genome, but also among the *BDG* genes. Future studies will be conducted to confirm the role of *BDG* genes in cuticle biosynthesis and, furthermore, their implication in resistance to biotic/abiotic stress. In addition, the data generated by the whole genome resequencing of these genotypes will allow for the analysis of additional genes which, if found to be useful, could be introgressed from the wild type into future varieties in goji berry breeding programs in Romania.

## Figures and Tables

**Figure 1 ijms-25-02130-f001:**
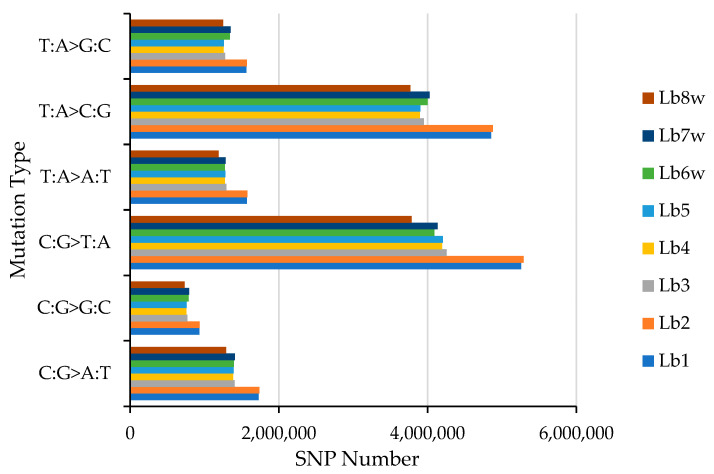
SNP mutation type distribution. SNP—Single Nucleotide Polymorphism, T—Thymine, A—Adenine, C—Cytosine, G—Guanine; Lb1–Lb8w are the tested genotypes.

**Figure 2 ijms-25-02130-f002:**
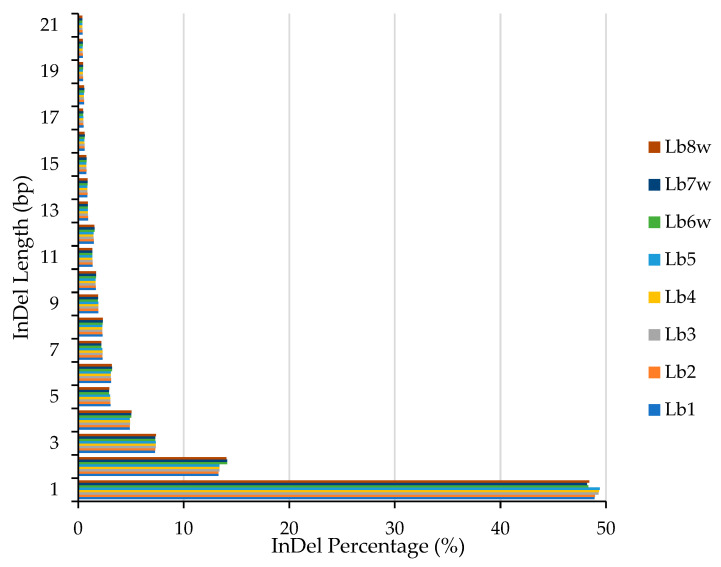
Length distribution of InDels in the eight Romanian goji berry genomes. Lb1–Lb8w are the tested genotypes.

**Figure 3 ijms-25-02130-f003:**
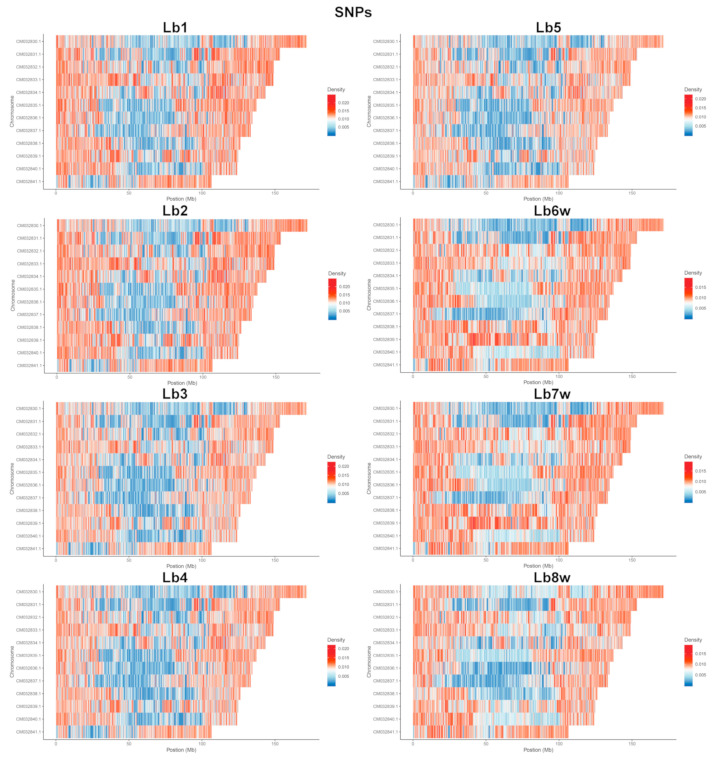
SNP densities per chromosome, per genotype. SNP—Single Nucleotide Polymorphism; Lb1–Lb8w are the tested genotypes.

**Figure 4 ijms-25-02130-f004:**
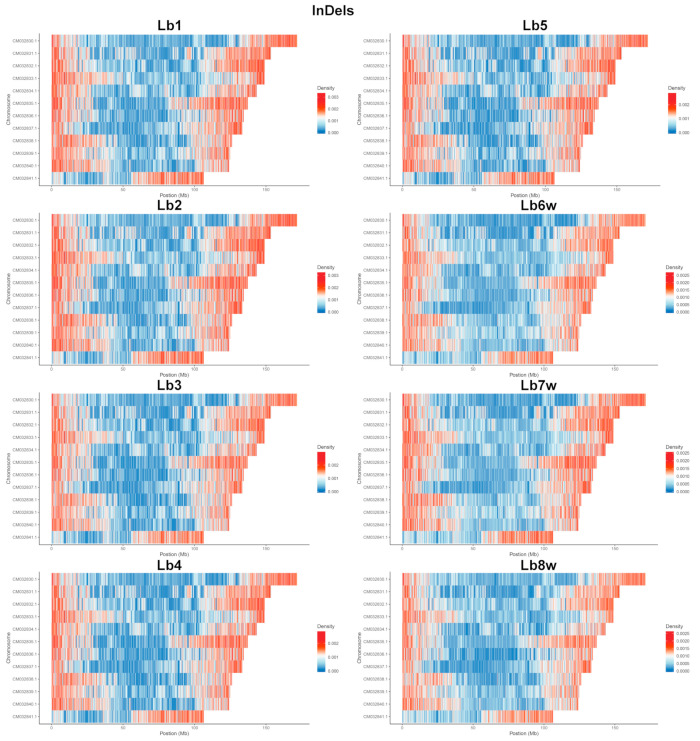
InDel densities per chromosome, per genotype. InDels—insertions and deletions; Lb1–Lb8w are the tested genotypes.

**Figure 5 ijms-25-02130-f005:**
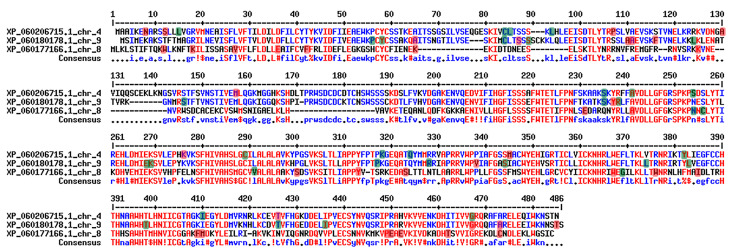
Alignment of the three BDG proteins. Highlighted with red are the non-synonymous SNPs, with green are the synonymous SNPs. Red colour fonts denote high consensus, blue colour fonts denote low consensus, and black colour fonts denote no consensus.

**Table 1 ijms-25-02130-t001:** Name, description, and location of the *BODYGUARD* gene in the *L. barbarum* and *L. feroscissimum* genomes.

Name/Gene ID	Description	Location
LOC132634709 ID: 132634709	probable lysophospholipase BODYGUARD 3 [*Lycium barbarum* (goji berry)]	Chromosome 4, NC_083340.1
LOC132607278 ID: 132607278	probable lysophospholipase BODYGUARD 4 [*Lycium barbarum* (goji berry)]	Chromosome 8, NC_083344.1
LOC132609965 ID: 132609965	probable lysophospholipase BODYGUARD 3 [*Lycium barbarum* (goji berry)]	Chromosome 9, NC_083345.1
LOC132060388 ID: 132060388	probable lysophospholipase BODYGUARD 3 [*Lycium ferocissimum*]	Chromosome 1, NC_081342.1
LOC132049371 ID: 132049371	probable lysophospholipase BODYGUARD 4 [*Lycium ferocissimum*]	Chromosome 3, NC_081344.1
LOC132030714 ID: 132030714	probable lysophospholipase BODYGUARD 3 [*Lycium ferocissimum*]	Chromosome 9, NC_081350.1

**Table 2 ijms-25-02130-t002:** Sequence analysis of goji berry *BDG* gene, LOC132634709, on chromosome 4.

Nr. crt.	SNP Position in Coding Sequence	Codon	Amino Acid	Reference Genome ASM1917538v2	Cultivated Genotypes					Wild Genotypes		
					Lb1	Lb2	Lb3	Lb4	Lb5	Lb6w	Lb7w	Lb8w
1	517	AAT/AGT	Asn/Ser	Asn	h69	h59	h67	h50	h75	Asn	Asn	Asn
2	529	AGC/ATC	Ser/Ile	Ser	h69	h57	h40	h50	h75	Ser	Ser	Ser
3	537	TTA/CTA	Leu/Leu (silent)	Leu (TTA)	Leu (CTA)	Leu (CTA)	Leu (CTA)	Leu (CTA)	Leu (CTA)	h33	h20	h30
4	590	CTT/CTC	Leu/Leu (silent)	Leu (CTT)	Leu (CTC)	Leu (CTC)	Leu (CTC)	Leu (CTC)	Leu (CTC)	h37	Leu (CTT)	h40
5	746	TGC/TGT	Cys/Cys (silent)	Cys (TGC)	Cys (TGT)	Cys (TGT)	Cys (TGT)	Cys (TGT)	Cys (TGT)	h50	h17	h30
6	749	CTG/CTC	Leu/Leu (silent)	Leu (CTG)	h35	h40	h50	h33	h20	h43	h29	h36
7	767	CTA/CTG	Leu/Leu (silent)	Leu (CTA)	Leu (CTG)	Leu (CTG)	Leu (CTG)	Leu (CTG)	Leu (CTG)	h50	h37	h30
8	804	CCA/ACA	Pro/Thr	Pro	h78	h45	h25	h80	h60	Pro	Pro	Pro
9	873	GCT/TCT	Ala/Ser	Ala	h67	h56	h18	h60	h80	Ala	Ala	Ala
10	956	ATG/ATT	Met/Ile	Met	h42	h35	h37	h40	h80	Met	Met	Met
11	1190	TCA/TCT	Ser/Ser (silent)	Ser (TCA)	h46	h92	h50	h45	h60	Ser (TCA)	Ser (TCA)	Ser (TCA)
12	1208	GCA/GCC	Ala/Ala (silent)	Ala (GCA)	h50	h7	h50	h60	h40	h75	h40	h33
13	1247	AGT/AGC	Ser/Ser (silent)	Ser (AGT)	h50	h20	h57	h70	h25	h80	h45	h40
14	1312	AAA/AGA	Lys/Arg	Lys	Arg	Arg	Arg	Arg	Arg	h80	h40	h50
15	1352	TGC/TGT	Cys/Cys (silent)	Cys(TGC)	h42	h39	h53	h57	h44	Cys (TGC)	Cys (TGC)	h14
16	1439	AAA/AAG	Lys/Lys (silent)	Gly	h59	h65	h60	h18	h60	h67	h57	h57
17	1457	CAG/CAA	Gln/Gln (silent)	Gln (CAG)	Gln (CAG)	Gln (CAG)	Gln (CAG)	Gln (CAG)	Gln (CAG)	h67	h57	h57
18	1520	ATG/ATT	Met/Ile	Met	h43	h54	h67	h83	h45	Met	Met	Met
19	1631	TAC/TAT	Tyr/Tyr (silent)	Tyr (TAC)	Tyr (TAT)	Tyr (TAT)	Tyr (TAT)	Tyr (TAT)	Tyr (TAT)	h75	h25	h73
20	1715	ATA/ATT	Ile/Ile (silent)	Ile	h61	h47	h50	h50	h50	h67	h33	h60
21	1761	ACG/TCG	Thr/Ser	Tyr	h67	h71	h50	h37	h33	Tyr	Tyr	Tyr
22	1886	GGC/GGG	Gly/Gly (silent)	Gly (GGC)	Gly (GGC)	h36	h54	h86	h30	h83	h43	h75

SNP—Single Nucleotide Polymorphism, T—Thymine, A—Adenine, C—Cytosine, G—Guanine; Lb1–Lb8w are the tested genotypes. Cells highlighted with blue are the polar amino acids, green cells are the nonpolar amino acids, and yellow cells are the basic amino acids.

**Table 3 ijms-25-02130-t003:** Sequence analysis of goji berry *BDG* gene, LOC132607278, on chromosome 8.

Nr. crt.	SNP Position in Coding Sequence	Codon	Amino Acid	Reference Genome ASM1917538v2	Cultivated Genotypes					Wild Genotypes		
					Lb1	Lb2	Lb3	Lb4	Lb5	Lb6w	Lb7w	Lb8w
1	129	TGG/TTG	Trp/Leu	Trp	h50	h67	h67	h57	h50	h33	h75	h75
2	146	AAA/GAA	Lys/Glu	Lys	Glu	Glu	Glu	Glu	Glu	Glu	Glu	Glu
3	174	GTA/GCA	Val/Ala	Val	h45	h58	h17	h33	h71	Val	Val	Val
4	202	GAG/GAC	Glu/Asp	Glu	Asp	h93	Asp	Asp	Asp	h41	h62	Asp
5	209	TTT/CTT	Phe/Leu	Phe	Phe	Phe	Phe	Phe	Phe	h46	h62	h67
6	287-288	AAA/CGA	Lys/Arg	Lys	h50	h47	h20	h50	h82	Lys	h14/Lys	Lys
7	392-394	AAA/GAC	Lys/Asp	Lys	h35	h53	h33	h33	h67	Lys	Lys	Lys
8	617	GAA/AAA	Glu/Lys	Glu	Glu	Glu	Glu	Glu	Glu	h57	h75	h75
9	648	GCA/GGA	Ala/Gly	Ala	Ala	Ala	Ala	Ala	Ala	h57	h12	h14
10	688	AAC/AAT	Asn/Asn (sIlent)	Asn	h41	h36	h57	h50	h25	Asn	Asn	Asn
11	694	TGC/TGT	Cys/Cys (sIlent)	Cys	h41	h36	h57	h43	h14	Cys	Cys	Cys
12	799	GTA/GTG	Val/Val (sIlent)	Val(GTA)	Val(GTA)	Val(GTA)	Val(GTA)	Val(GTA)	Val(GTA)	h50	h40	Val(GTG)
13	824	TCT/CCT	Ser/Pro	Pro	Pro	Pro	Pro	Pro	Pro	h43	h45	Pro
14	864	TAC/TTC	Tyr/Phe	Tyr	Tyr	Tyr	Tyr	Tyr	Tyr	h50	h33	Tyr
15	891	AGT/ATT	Ser/Ile	Ser	h53	h58	h56	h78	h67	Ser	Ser	Ser
16	1023	TGG/TTG	Trp/Leu	Trp	h6	Trp	Trp	Trp	Trp	h50	h71	h43
17	1030	GGA/GGT	Gly/Gly (sIlent)	Gly (GGA)	h58	h22	h43	h56	h33	Gly (GGA)	Gly (GGA)	Gly (GGA)
18	1049	TGG/GGG	Trp/Gly	Trp	h38	h78	h43	h44	h71	Trp	Trp	Trp
19	1076	ATT/GTT	Ile/Val	Ile	Ile	Ile	Ile	Ile	Ile	h40	h50	h67
20	1150	TTT/TTA	Phe/Leu	Phe	Phe	Phe	Phe	Phe	Phe	h29	h50	h50
21	1151	ATG/GTG	Met/Val	Met	Met	Met	Met	Met	Met	h29	h50	h50
22	1266	CCT/CTT	Pro/Leu	Pro	h64	h75	h60	h64	h50	Pro	Pro	Pro
23	1270	GAA/GAT	Glu/Asp	Glu	Asp	Asp	Asp	Asp	Asp	h60	h62	Asp
24	1274	GAG/AAG	Glu/Lys	Glu	Glu	Glu	Glu	Glu	Glu	h60	h57	Lys
25	1304	ACT/TCT	Thr/Ser	Thr	h64	h73	h71	h86	h40	Thr	Thr	Thr
26	1338	TGT/TCT	Cys/Ser	Cys	h36	h23	h37	Ser	h71	Cys	Cys	Cys

SNP—Single Nucleotide Polymorphism, T—Thymine, A—Adenine, C—Cytosine, G—Guanine; Lb1–Lb8w are the tested genotypes. Cells highlighted with blue are the polar amino acids, green cells are the nonpolar amino acids, pink cells are the acidic amino acids, and yellow cells are the basic amino acids.

**Table 4 ijms-25-02130-t004:** Sequence analysis of goji berry *BDG* gene, LOC132609965, on chromosome 9.

Nr. crt.	SNP Position in Coding Sequence	Codon	Amino Acid	Reference Genome ASM1917538v2	Cultivated Genotypes					Wild Genotypes		
					Lb1	Lb2	Lb3	Lb4	Lb5	Lb6w	Lb7w	Lb8w
1	473	CCT/CCC	Pro/Pro (silent)	Pro	Pro	Pro	Pro	Pro	Pro	h75	h56	Pro
2	479	TAC/TAT	Tyr/Tyr (silent)	Tyr (TAC)	Tyr(TAT)	Tyr(TAT)	Tyr(TAT)	Tyr(TAT)	Tyr(TAT)	h50	h44	h33
3	498	GCC/ACC	Ala/Thr	Ala	h61	h41	h67	h67	h17	h40	h50	Ala
4	548	TGT/TGC	Cys/Cys (silent)	Cys (TGT)	Cys (TGT)	Cys (TGT)	Cys (TGT)	Cys (TGT)	Cys (TGT)	h50	h54	Cys (TGT)
5	555-557	TCT/GCT/TCA/TCG	Ser/Ala/Ser/Ser	Ser	Ala	Ala	Ala	Ala	Ala	h50	h46	h33
6	563	TCT/TCC	Ser/Ser (silent)	Ser (TCT)	Ser(TCC)	Ser(TCC)	Ser(TCC)	Ser(TCC)	Ser(TCC)	h67	h46	Ser(TCT)
7	625	GCG/GTG	Ala/Val	Ala	Val	Val	Val	Val	Val	Ala	h70	h40
8	627	GCT/TCT	Ala/Ser	Ala	Ser	Ser	Ser	Ser	Ser	Ser	Ser	h40
9	643	TTC/TCC	Phe/Ser	Phe	Ser	Ser	Ser	Ser	Ser	Ser	Ser	Ser
10	668	CTT/CTC	Leu/Leu (silent)	Leu (CTT)	Leu(CTC)	Leu(CTC)	Leu(CTC)	Leu(CTC)	Leu(CTC)	h33	h60	h50
11	713	TCG/TCC	Ser/Ser (silent)	Ser (TCG)	Ser(TCC)	Ser(TCC)	Ser(TCC)	Ser(TCC)	Ser(TCC)	h25	h33	h75
12	977	TCG/TCA	Ser/Ser (silent)	Ser (TCG)	Ser (TCA)	Ser (TCA)	Ser (TCA)	Ser (TCA)	Ser (TCA)	h33	h20	h60
13	982	TAT/TGT	Tyr/Ser	Tyr	h50	h67	h50	h50	h50	Tyr	Tyr	Tyr
14	986	CGG/CGA	Arg/Arg (silent)	Arg (CGG)	Arg (CGA)	Arg (CGA)	Arg (CGA)	Arg (CGA)	Arg (CGA)	h33	h20	h63
15	1076	GAG/GAA	Glu/Glu (silent)	Glu(GAG)	Glu (GAA)	Glu (GAA)	Glu (GAA)	Glu (GAA)	Glu (GAA)	h33	h20	h70
16	1079	AAA/AAG	Lys/Lys (silent)	Lys (AAA)	h58	h47	h40	h43	h67	h25	h25	h70
17	1223	CCA/CCC	Pro/Pro (silent)	Pro (CCA)	h50	h48	h22	h60	h57	Pro (CCA)	Pro (CCA)	Pro (CCA)
18	1256	AGG/AGA	Arg/Arg (silent)	Arg (AGG)	h41	h56	h71	h56	h50	h37	h67	Arg(AGA)
19	1285	GTG/GCG	Val/Ala	Val	h42	h44	h67	h60	h62	h37	h775	Ala
20	1304	TCG/TCT	Ser/Ser (silent)	Ser(TCG)	h65	h48	h17	h50	h33	Ser (TCG)	Ser (TCG)	Ser (TCG)
21	1394	CTG/CTC	Leu/Leu (silent)	Leu(CTG)	Leu(CTC)	Leu(CTC)	Leu(CTC)	Leu(CTC)	Leu(CTC)	h50	h50	Leu(CTC)
22	1421	TTA/TTG	Leu/Leu (silent)	Leu (TTA)	h67	h62	h60	Leu (TTG)	Leu (TTA)	Leu (TTA)	Leu (TTA)	Leu (TTA)
23	1466	ACT/ACA	Thr/Thr (silent)	Thr(ACT)	Thr (ACA)	Thr (ACA)	Thr (ACA)	Thr (ACA)	Thr (ACA)	h50	h40	h57
24	1550	ACA/ACG	Thr/Thr (silent)	Thr(ACA)	Thr (ACG)	Thr (ACG)	Thr (ACG)	Thr (ACG)	Thr (ACG)	h33	h50	h57
25	1580	ATC/ATA	Ile/Ile (silent)	Ile (ATC)	Ile (ATA)	Ile (ATA)	Ile (ATA)	Ile (ATA)	Ile (ATA)	h33	h57	h71
26	1671-1672	GGA/AGA/GAA	Gly/Arg/Glu	Gly	h16	h9	h16	h11	h6	Gly	Gly	Gly
27	1689	GCT/TCT	Ala/Ser	Ala	h23	h15	h21	h20	h24	Ala	Ala	Ala
28	1726	ACA/AAA	Thr/Lys	Thr	h10	h17	h33	h18	h23	h57	h17	h67

SNP—Single Nucleotide Polymorphism, T—Thymine, A—Adenine, C—Cytosine, G—Guanine; Lb1–Lb8w are the tested genotypes. Cells highlighted with blue are the polar amino acids, green cells are the nonpolar amino acids, pink cells are the acidic amino acids, and yellow cells are the basic amino acids.

## Data Availability

Data are contained within the article and Appendix A.

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
