# Peer review of "Genomic Analysis of Romanian Lycium Genotypes: Exploring BODYGUARD Genes for Stress Resistance Breeding"

_ijms, 2024, doi:10.3390/ijms25042130_

Round 1

Reviewer 1 Report (New Reviewer)

Comments and Suggestions for Authors

Overall, this research article entitled “Genomic Analysis of Romanian Lycium Genotypes: Exploring BODYGUARD Genes for Stress Resistance Breeding”. represents an interesting investigation focusing on the genetic analysis of Romanian Lycium (goji berry) genotypes. The research aims to explore the genetic variation at the whole genome level, particularly focusing on the BODYGUARD (BDG) genes related to cuticle resistance. Abstract is logical providing the concise summary of the findings, The introduction provides sufficient background, the methods are generally appropriate for the experiments conducted. The analysis and results presented in figures seem logical while interpretation is supported by results. Moreover, the results are clearly described making the manuscript understandable for readers. In order to improve the present study, some essential modifications have to be fixed before it proceeds, and decisive action can be taken. In addition, the study needs some editing on some minor grammatical issues. All the comments and remarks are given below.

The objective of exploring BDG genes in Romanian goji berries for stress resistance breeding is well-defined and relevant to agriculture and plant breeding. The study's focus on local varieties and their potential for enhancing stress resistance is commendable.

The methods used for genomic analysis, including whole-genome sequencing and SNP (Single Nucleotide Polymorphisms) analysis, are appropriate and well-executed. However, it would benefit from a more detailed description of the quality control measures employed during genomic sequencing. How were errors minimized during the sequencing process, and what validation steps were taken?

The results indicating significant differences between cultivated and wild genotypes at the SNP level are intriguing. The study successfully identifies key SNPs and InDels differences, which is crucial for breeding purposes. Nonetheless, a more thorough discussion on how these genetic variations translate to phenotypic traits or stress resistance would be beneficial.

The discussion effectively ties the findings to the broader context of goji berry breeding and genetic diversity. However, the study could benefit from a comparison with similar studies in other regions or species to position its findings within the global research landscape.

The manuscript is generally well-written and structured. However, careful proofreading to correct minor grammatical errors and improve sentence structures could enhance its readability. Some examples are given below:

In abstract in L15: “using as planting material non-native genotypes” hard to understand this sentence.

In L16-18: “By 2023, in Romania were successfully registered seven varieties of goji berries, developed using germplasm that originated from sources outside the country”. Consider rewriting the sentence as: By 2023, seven varieties of goji berries were successfully registered in Romania, developed using germplasm that originated from sources outside the country.

In L26: “The research uncovered also significant SNPs and InDels” should be, The research also uncovered significant SNPs and InDels.

L47: Yao and al. (2018)??? Should be Yao et al.

Overall, this manuscript makes a valuable contribution to the field of plant genomics and breeding, particularly in the context of Romanian Lycium genotypes. With some improvements in detailing the methodology, expanding the discussion, and refining the presentation, it could be a significant resource for researchers in this field.

Comments on the Quality of English Language

The manuscript is generally well-written and structured. However, careful proofreading to correct minor grammatical errors and improve sentence structures could enhance its readability. Some examples are given below:

In abstract in L15: “using as planting material non-native genotypes” hard to understand this sentence.

In L16-18: “By 2023, in Romania were successfully registered seven varieties of goji berries, developed using germplasm that originated from sources outside the country”. Consider rewriting the sentence as: By 2023, seven varieties of goji berries were successfully registered in Romania, developed using germplasm that originated from sources outside the country.

In L26: “The research uncovered also significant SNPs and InDels” should be, The research also uncovered significant SNPs and InDels.

L47: Yao and al. (2018)??? Should be Yao et al.

Author Response

Response to Reviewer 1 Comments

1. Summary

Thank you very much for taking the time to review our manuscript. Please find the detailed responses below and the corresponding revisions/corrections highlighted/in track changes in the re-submitted files.

2. Questions for General Evaluation

Reviewer’s Evaluation

Response and Revisions

Does the introduction provide sufficient background and include all relevant references?

Yes

Are all the cited references relevant to the research?

Yes

Is the research design appropriate?

Yes

Are the methods adequately described?

Can be improved

We have improved the material and methods

Are the results clearly presented?

Can be improved

We extended the results presentation

Are the conclusions supported by the results?

Yes

3. Point-by-point response to Comments and Suggestions for Authors

Comments 1: In abstract in L15: “using as planting material non-native genotypes” hard to understand this sentence.

Response 1: Thank you for pointing this out. We have changed the entire sentence. The new form is ` Because of growing demand, Europe and North America are increasing their goji berry production, using goji berry varieties that are not originally from these regions.`

Comments 2: In L16-18: “By 2023, in Romania were successfully registered seven varieties of goji berries, developed using germplasm that originated from sources outside the country”. Consider rewriting the sentence as: By 2023, seven varieties of goji berries were successfully registered in Romania, developed using germplasm that originated from sources outside the country.

Response 2: Thank you for this suggestion. We have made the change accordingly.

Comments 3: In L26: “The research uncovered also significant SNPs and InDels” should be, The research also uncovered significant SNPs and InDels.

Response 3: Thank you for this suggestion. We have made the change accordingly.

Comments 4: L47: Yao and al. (2018)??? Should be Yao et al.

Response 4: Thank you for this suggestion. We have made the change accordingly.

Comments 5: The methods used for genomic analysis would benefit from a more detailed description of the quality control measures employed during genomic sequencing. How were errors minimized during the sequencing process, and what validation steps were taken?

Response 5: Following your suggestion, we have made the following corrections:
-In Materials and Methods section, Sequencing subsection, we have added `and Sequencing Data Quality Control`.

-The paragraph about Quality of the analysis from 4.4 was moved to 4.3 and further details were given.

-In the Results section, subsection 2.1.1. Sequencing Data Quality Control, we have added info about Sequencing Quality Distribution, Sequencing Error Rate, Sequencing Data Filtration and Statistics of Sequencing Data.

Comments 6: A more thorough discussion on how SNP genetic variations translate to phenotypic traits or stress resistance would be beneficial need to be provided.

Response 6: We added two paragraphs in the discussion section, one about studies that used the Arabidopsis bdg mutant phenotype to prove BDG1 functions, and one about how these SNP genetic variations could translate into phenotypical differences between the wild type and cultivated plants.

Comments 7: A comparison with similar studies in other regions or species to position its findings within the global research landscape

Response 7: We have included one paragraph in the Discussion section, that illustrate how advancements in Genetic research can be used to enhance crop breeding in general and advance de genetic research and breeding in particular for goji berry.

4. Response to Comments on the Quality of English Language

Point 1: Moderate editing of English language required.

Response 1: We reviewed the English language.

5. Additional clarifications

Reviewer 2 Report (New Reviewer)

Comments and Suggestions for Authors

The present manuscript entitled “Genomic Analysis of Romanian Lycium Genotypes: Exploring BODYGUARD Genes for Stress Resistance Breeding” demonstrated the difference between the wild and cultivated goji berry in molecular level. Results suggested significant SNPs and InDels differences between cultivated and wild genotypes, in the entire genome, providing crucial insights for goji berry breeders to support the development of goji berry cultivation in Romania. The manuscript is nicely written and presented, although there are some small suggestions for further improvement, please find them below.

1.      Stress resistance genes are very broad term, authors may specify the type of stress which has been of major interest in the title.

2.      Abstract and introduction is quite informative and well organized. Although line 100-120 text may be more suitable for discussion section.

3.      Table 1 contains the information about 6 genomes; however, the study was done in 8 genomes. Please check if some information is missing.

4.      Line 315-317, the 7 varieties were created in Romania, please include more details and reference for this claim.

Thank you

Author Response

Response to Reviewer 2 Comments

1. Summary

2. Questions for General Evaluation

Reviewer’s Evaluation

Response and Revisions

Does the introduction provide sufficient background and include all relevant references?

Can be improved

We have improved the introduction

Are all the cited references relevant to the research?

Yes

Is the research design appropriate?

Yes

Are the methods adequately described?

Yes

Are the results clearly presented?

Yes

Are the conclusions supported by the results?

Yes

3. Point-by-point response to Comments and Suggestions for Authors

Comments 1: Stress resistance genes are very broad term, authors may specify the type of stress which has been of major interest in the title.

Response 1: Thank you for pointing this out. As the study is at its beginning stage, and the BDG gene is involved in cuticle synthesis, that play a role in both biotic and abiotic resistance, we choose to keep the title shorter. It is indeed true that the analysis came in the context of a project related to resistance to eriophyid mites, but the findings may be related to so many other types of resistance, that we chose not to be more specific.

Comments 2: Abstract and introduction is quite informative and well organized. Although line 100-120 text may be more suitable for discussion section.

Response 2: Thank you for your suggestion. The paragraph was moved at the beginning of Discussion section.

Comments 3: Table 1 contains the information about 6 genomes; however, the study was done in 8 genomes. Please check if some information is missing.

Response 3: Table 1 contains the information about 6 genes, more exactly it gives information on name, description, and location of the BODYGUARD genes in the L. barbarum and L. feroscissimum genomes.

Comments 4: Line 315-317, the 7 varieties were created in Romania, please include more details and reference for this claim

Response 2: We have added a new paragraph regarding the 7 varieties created in Romania, and the reference where these seven varieties are described, with pictures included. (Acta Hortic. 1381. ISHS 2023. DOI 10.17660/ActaHortic.2023.1381.49)

This manuscript is a resubmission of an earlier submission. The following is a list of the peer review reports and author responses from that submission.

Round 1

Reviewer 1 Report

Comments and Suggestions for Authors

1. The title needs to be rewritten to accurately reflect the main research content of this article.

2. The current materials used are insufficient to fully describe the genetic diversity of Romanian Lycium species.

3. The introduction section needs further refinement to make it more concise.

4. It is recommended to include the main content conducted by the sequencing biotech company as an attachment.

5. There are additional genetic diversity indicators that need to be presented and analyzed.

6. Detailed descriptions of the characteristics of the varieties, as well as information about the sampling locations, time, and methods, need to be provided.

7. Accurate species identification is necessary, including information on the storage of voucher specimens.

8. The main analysis section needs to be strengthened.

9. In the discussion section, the experimental data should be analyzed and discussed based on existing research literature.

10. It is advisable to analyze and argue the results of this study in detail after reviewing the global molecular genetic diversity of Lycium.

Comments on the Quality of English Language

Moderate editing of English language required.

Author Response

For research article `Exploring Genetic Diversity in Romanian Lycium Species: A Comparative Study of Wild and Cultivated Genotypes`

BODYGUARD genes of Wild Romanian Lycium Genotypes as potential sources of stress resistance for Romanian breeding

Comparative Study of

Exploring Genetic Diversity in: A

Response to Reviewer 1 Comments

1. Summary

2. Questions for General Evaluation

Reviewer’s Evaluation

Response and Revisions

Does the introduction provide sufficient background and include all relevant references?

Can be improved

We have improved the introduction, please see the Word file with track changes or the PDF, in final form.

Are all the cited references relevant to the research?

Yes

Is the research design appropriate?

Yes

Are the methods adequately described?

Yes

Are the results clearly presented?

Must be improved

We extended the results presentation

Are the conclusions supported by the results?

Yes

3. Point-by-point response to Comments and Suggestions for Authors

Comments 1: The title needs to be rewritten to accurately reflect the main research content of this article.

Response 1: Thank you for pointing this out. We have changed the title to reflect better the content of the article. The new title is ”Exploring BODYGUARD Genes in Wild Romanian Lycium Genotypes as Potential Sources of Enhanced Stress Resistance for Goji Berry Breeding Programs”

Comments 2: The current materials used are insufficient to fully describe the genetic diversity of Romanian Lycium species.

Response 2: We realized from your previous comment that the title was misleading. Our intent was not to make a comprehensive study on Romanian Lycium species, but to illustrate the fact that the wild genotypes could be used as a valuable source of resistance traits.

Comments 3: The introduction section needs further refinement to make it more concise.

Response 3: We refined the introduction.

Comments 4: It is recommended to include the main content conducted by the sequencing biotech company as an attachment.

Response 4: We attached the Supplementary Files 1 and 2, containing the Novogene sequencing data used in the article.

Comments 5: There are additional genetic diversity indicators that need to be presented and analyzed.

Response 5: As mentioned at the response 2, our aim was not to characterize the Romanian Lycium genetic diversity. Thank you once again for pointing this out.

Comments 6: Detailed descriptions of the characteristics of the varieties, as well as information about the sampling locations, time, and methods, need to be provided.

Response 6: We add in the Supplementary file 1 the metadata about the sampling. For characteristics of the genotypes, the detailed morphological description is available in the cited article no 31, that was published within the project.        The article is ”Luchian, V.; Ciceoi, R.; Gutue, M. Comparative Leaf and Flower Morpho-Anatomical Study of Wild and Cultivated Gojiberry (Lycium Barbarum L.) in Romania. Sci. Pap. Ser. B Hortic. 2022, LXVI”.

Comments 7: Accurate species identification is necessary, including information on the storage of voucher specimens.

Response 7: All the genotypes used in the study are L. barbarum. The species identification was done when the specimens were added in the USAMV Bucharest Herbarium. We added this information at Materials and Methods. Thank you for the valuable comment.   

Comments 8: The main analysis section needs to be strengthened.

Response 8: We added more details in both Results and Discussions sections.

Comments 9: In the discussion section, the experimental data should be analyzed and discussed based on existing research literature.

Response 9: We made additional discussions based on existing published studies.

Comments 10: It is advisable to analyze and argue the results of this study in detail after reviewing the global molecular genetic diversity of Lycium.

Response 10: We analyzed and argued the results regarding the whole genome sequencing and the BDG genes analysis case study.

In our present work, we did not used molecular markers, such as SSR, ISSR, SRAP, RAPD, etc., that are usually used for this type of studies.  

4. Response to Comments on the Quality of English Language

Point 1: Moderate editing of English language required.

Response 1: We reviewed the English language.

5. Additional clarifications

Reviewer 2 Report

Comments and Suggestions for Authors

The reason for this decision is:

This manuscript does not fulfill the standards established for the journal to be considered for publication.

It seems like a good study was done on the diversity of genetic resources with 100 species of Wolfberry. However, considering the characteristics of this journal, it cannot be reviewed, and I would like to recommend it to another sister journal.

Author Response

For research article `Exploring Genetic Diversity in Romanian Lycium Species: A Comparative Study of Wild and Cultivated Genotypes`

BODYGUARD genes of Wild Romanian Lycium Genotypes as potential sources of stress resistance for Romanian breeding

Comparative Study of

Exploring Genetic Diversity in: A

Response to Reviewer 2 Comments

1. Summary

2. Questions for General Evaluation

Reviewer’s Evaluation

Response and Revisions

Does the introduction provide sufficient background and include all relevant references?

Can be improved

We have improved the introduction

Are all the cited references relevant to the research?

Can be improved

We have improved the references, by adding new relevant ones

Is the research design appropriate?

Can be improved

We have improved the presentation of the research design

Are the methods adequately described?

Can be improved

We have improved the methods

Are the results clearly presented?

Can be improved

We have improved the results presentation

Are the conclusions supported by the results?

Can be improved

We have improved the conclusions

3. Point-by-point response to Comments and Suggestions for Authors

Comments 1: This manuscript does not fulfill the standards established for the journal to be considered for publication.

It seems like a good study was done on the diversity of genetic resources with 100 species of Wolfberry. However, considering the characteristics of this journal, it cannot be reviewed, and I would like to recommend it to another sister journal.

Response 1: Thank you for pointing this out. We have changed the title to reflect better the content of the article. We agree that it was misleading.

The new title is ”Exploring BODYGUARD Genes in Wild Romanian Lycium Genotypes as Potential Sources of Enhanced Stress Resistance for Goji Berry Breeding Programs”.

The reviewed manuscript with track changes was uploaded, so that all the corrections are visible.

Reviewer 3 Report

Comments and Suggestions for Authors

Dear Authors,

Review of the manuscript titled: Exploring Genetic Diversity in Romanian Lycium Species: A Comparative Study of Wild and Cultivated Genotypes. Authors: Roxana Ciceoi, Adrian Asănică, Vasilica Luchian and Mihaela Iordăchescu.

The manuscript presents a comprehensive exploration into the genetic diversity of Romanian Lycium species, focusing on both cultivated and wild genotypes. The study delves into the context of the global market demand for goji berries and the efforts to expand local production in America and the Europe, particularly in Romania.

The authors effectively introduce the subject matter, providing a detailed overview of the background, emphasizing the significance of understanding genetic variations in Lycium species, especially concerning the market demand, local adaptation, and response to climate change. The methodology employed in this research is well-documented and adequately presented, ensuring transparency in the study's approach.

While the initial sections of the manuscript are commendable, there are areas that require enhancement. The presentation of results could benefit from improvement, as it lacks clarity in effectively conveying the findings obtained from the advanced genomic studies. However, the authors redeem the manuscript through a satisfactory discussion section. They adeptly analyze their research findings in conjunction with available literature sources, offering insightful interpretations and implications for goji berry breeders.

Moreover, the manuscript marks the beginning of a broader project that holds promise for investigating genes associated with resistance to various stresses, particularly focusing on cuticle thickness. The authors rightly highlight the importance of this aspect in ensuring plant survival during severe droughts, insolation, and pest attacks. The exploration of BODYGUARD genes and their role in plant defense mechanisms against diverse stressors signifies a valuable contribution to the field of goji berry cultivation.

 My comments that will help improve this manuscript are below:

1) Abstract - it must be thoroughly redacted, the part concerning the introduction to the research topic is too extensive, and the methodology and research results are presented too briefly. No research hypothesis.

2) Please arrange the keywords in alphabetical order.

3) In the paragraph from Line135 to Line 144 - there is no clearly defined research hypothesis. And the purpose of the research is too general.

4) Figure 1 - please explain all abbreviations used in the figure (write in general what the letters T, A, G and C mean, what the abbreviation SNP stands for, and write that Lb8w to Lb1 are the tested genotypes). Please take into account that figures and tables must be clear and understandable so that there is no need to look for explanations of abbreviations in the body of the manuscript.

5) Figure 2 - Y axis - Length of what? X-axis -Percentage of what? Explain that Lb8w to Lb1 are the genotypes being tested.

6) Figure 3 - is completely illegible. Please increase the font size.

7) Line 229 - the scientific name should be written in italics.

8) Tables 2, 3, and 4 - the legend is incomprehensible. Above "Cultivated" and "Wild" add "Genotepes". Explain the SNP.

9) Figure 4 - the photos are of poor quality, nothing specific is visible on them, they do not provide any information. Please change them.

In conclusion, the research conducted in this manuscript is robust and sets the stage for further investigations. While the presentation of results could be improved, the authors demonstrate a satisfactory level of analysis and discussion. The findings hold significance for goji berry breeders in Romania, offering valuable insights into genetic diversity and stress resistance mechanisms in Lycium species.

To sum up, I believe that the Dear Editors of the IJMS journal should consider publishing this manuscript.

Author Response

For research article `Exploring Genetic Diversity in Romanian Lycium Species: A Comparative Study of Wild and Cultivated Genotypes`

BODYGUARD genes of Wild Romanian Lycium Genotypes as potential sources of stress resistance for Romanian breeding

Comparative Study of

Exploring Genetic Diversity in: A

Response to Reviewer 3 Comments

1. Summary

2. Questions for General Evaluation

Reviewer’s Evaluation

Response and Revisions

Does the introduction provide sufficient background and include all relevant references?

Can be improved

The introduction was updated. Please see the modifications in the attached revised version

Are all the cited references relevant to the research?

Yes

Is the research design appropriate?

Yes

Are the methods adequately described?

Yes

Are the results clearly presented?

Must be improved

The Results section was updated. Please see the modifications in the attached revised version

Are the conclusions supported by the results?

Yes

3. Point-by-point response to Comments and Suggestions for Authors

Comments 1: Abstract - it must be thoroughly redacted, the part concerning the introduction to the research topic is too extensive, and the methodology and research results are presented too briefly. No research hypothesis.

Response 1: Thank you for pointing this out. We have updated the abstract, reducing the introductory part and focusing on the methodology and results. Therefore, we have re-written the abstract, the correction being visible with track changes, from line 16 to line 34. The Abstract now reads as follow: “[Goji berries, long valued in Chinese medicine and cuisine for their wide range of medicinal benefits, are now worldwide considered a 'superfruit' and functional food. Lycium barbarum L. and L. chinense Mill. currently dominate the market. Due to increasing demand, but also concerns about food safety and sustainability, Europe is expanding the local goji berry production, using as planting material Chinese genotypes. European breeding programs are focusing on Lycium breeding, in search of varieties adapted to local conditions and local market demands. By 2023, Romania registered seven goji berry varieties, but without incorporating local germplasm. A broader project focused on goji berry genotypes depicting resistance to abiotic and biotic stress, especially the goji berry pests, was initiated in 2021 in USAMV Bucharest. The current study analysis the genetic variation at the whole genome level and takes as a case study the differences in genomic coding sequences of BODYGUARD (BDG) 3 and 4 genes of chromosomes 4, 8, and 9, involved in cuticle related resistance. Five cultivated and three wild L. barbarum plant genomes have been used in this study. All three BDG genes show distinctive differences at SNP level be-tween the cultivated and wild type genotypes, especially for the genes present on chromosomes 4 and 8, higher than 64%. The research uncovered also significant SNPs and InDels differences be-tween cultivated and wild genotypes, in the entire genome, providing crucial insights for goji berry breeders to support the development of goji berry cultivation in Romania]”.

Comments 2: Please arrange the keywords in alphabetical order.

Response 2: Thank you for comment on this. We have updated the key words order, Therefore, the key-words, at line 30, now reads as follow: BODYGUARD genes, cuticle, goji berry breeding; plant resistance; Whole Genome Sequencing.

Comments 3: In the paragraph from Line 135 to Line 144 - there is no clearly defined research hypothesis. And the purpose of the research is too general.

Response 3: According to your comment, we have revised the last paragraph of the introduction, that is currently between lines 119-126. Although is it true that we did not have a defined research hypothesis for the presented research, as our research project was more directed through the goji berry resistance to Aceria kuko mites, we decided to explore more the findings revealed by morphological studies performed by Luchian et al, 2022 (reference 31), that showed that ”Notable morphological distinctions were observed in leaf shape, orientation, and width of Romanian L. barbarum” (line 325). We performed the whole genome sequencing as the basis for our exploration of resistance genes, because we have in our breeding sector both resistant and susceptible plants, and we hoped to find the answer by looking at the genes.   

Comments 4: Figure 1 - please explain all abbreviations used in the figure (write in general what the letters T, A, G and C mean, what the abbreviation SNP stands for, and write that Lb8w to Lb1 are the tested genotypes). Please take into account that figures and tables must be clear and understandable so that there is no need to look for explanations of abbreviations in the body of the manuscript.

Response 4: We have revised Figure 1 accordingly and explained the abbreviations.

Comments 5: Figure 2 - Y axis - Length of what? X-axis -Percentage of what? Explain that Lb8w to Lb1 are the genotypes being tested.

Response 5: We have revised Figure 2 accordingly and explained the X and Y axis.

Comments 6: Figure 3 - is completely illegible. Please increase the font size.

Response 6: We have revised Figure 3 accordingly, and split it into Figure 3 and Figure 4, to increase the readability for SNPs and InDels. The individual graphs can be also found in the Supplementary file 2.

Comments 7: Line 229 - the scientific name should be written in italics.

Response 7: Thank you very much for this observation. We corrected the Italics. With this occasion we noticed that there are differences between our uploaded files and the files we received back from IJMS. We will contact our editor.

Comments 8: Tables 2, 3, and 4 - the legend is incomprehensible. Above "Cultivated" and "Wild" add "Genotepes". Explain the SNP.

Response 8: We have revised the 3 tables and corrected the fonts and inserted the explanatory notes below the tables.

Comments 9: Figure 4 - the photos are of poor quality, nothing specific is visible on them, they do not provide any information. Please change them..

Response 9: We deleted the photos.

Comments 10: In conclusion, the research conducted in this manuscript is robust and sets the stage for further investigations. While the presentation of results could be improved, the authors demonstrate a satisfactory level of analysis and discussion. The findings hold significance for goji berry breeders in Romania, offering valuable insights into genetic diversity and stress resistance mechanisms in Lycium species.

To sum up, I believe that the Dear Editors of the IJMS journal should consider publishing this manuscript.

Response 10: Thank you very much for your recommendations.

Round 2

Reviewer 2 Report

Comments and Suggestions for Authors

This paper pointed out that the first revision contained errors that could not be corrected. In the second review, the English content in the text was also corrected, and there was no emergency regarding additional experiments. Numerous mechanical data confuse breeders and make the genetic classification of Romanian Lycium invalid. In particular, it is difficult to publish in this journal without research on the relationship between phenotype and gene sequence.